



# Atmospheric transmission patterns which promote persistent winter haze over Beijing

Muyuan Li[1,2], Yao Yao[1,2], Ian Simmonds[3], Dehai Luo[1,2], Linhao Zhong[1,2], Lin Pei[4]

[1]Key Laboratory of Regional Climate-Environment for Temperate East Asia, Institute of Atmospheric Physics, Chinese Academy of Sciences, Beijing, 100029, China

[2]University of Chinese Academy of Sciences, Beijing, 100049, China

[3]School of Earth Sciences, University of Melbourne, Parkville, Victoria, 3010, Australia

[4]Institute of Urban Meteorology, China Meteorological Administration, Beijing, 100089, China

*Correspondence to*: Yao Yao (yaoyao@tea.ac.cn)

**Abstract**. The persistent winter haze episodes that occurred in Beijing over the period 1980 to 2016 are examined based on both reanalysis and station data. On both interannual and intra-seasonal timescales, winter haze weather in Beijing is found to be associated with a significant atmospheric teleconnection pattern from the North Atlantic to Eurasia (Beijing). A positive North Atlantic Oscillation (NAO+) phase and a positive East Atlantic/West Russia (EA/WR+) phase can be observed as part of this teleconnection pattern (or an arched wave train). This study focuses mainly on the role of the NAO+ pattern, because the NAO index shows a closer relationship with winter haze frequency, especially after 1999, and the NAO+ pattern leads to the formation of persistent haze events over a longer period of time. Composite analyses show that a robust and consistent daily evolution of the wave train originates from an NAO+ pattern over the North Atlantic 8–10 days prior to the persistent haze events. The wave train continues propagating energy downstream, which leads to the formation and maintenance of a high-pressure center over northeast China, thus creating favorable meteorological conditions for the persistent haze events in Beijing. Thus, the NAO+ pattern is also an essential preceding background for the formation of the wave train, which can be treated as a potential predictor for persistent hazy weather. Corresponding to the NAO+ pattern, a tripolar sea surface



temperature mode and intensified zonal wind over the North Atlantic also serve as prior signals for the
persistent haze events. In addition, the propagation of the wave train is also associated with preceding
significant positive sea ice concentration anomalies in the Barents–Kara Sea. Moreover, comparative
analysis demonstrates that NAO+ winters are more advantageous to the formation and maintenance of
winter haze weather in Beijing rather than NAO− winters.
**1 Introduction**
Beijing, the capital city of China, is situated in the northeast of the country. It covers 16,410 km$^2$
and has a permanent population of 21.542 million. In 2019 Beijing's gross domestic product reached
3,537 billion yuan, an increase of 6.1 % over the previous year (http://www.stats.gov.cn). Along with
economic development, Beijing has had more frequent hazy weather, especially persistent haze in
winter, over the last 60 years (Wang et al., 2014; Su et al., 2015; Li et al., 2018a; Pei and Yan, 2018;
Pei et al., 2018; Shi et al., 2019). Haze pollution is associated with a high PM$_{2.5}$ concentration and low
visibility, and is harmful to human health (e.g., cardiovascular and respiratory diseases and lung cancer)
and puts pressure on public transportation and economic activities (Wang et al., 2013).
The serious impacts of haze pollution in Beijing have been the topic of numerous studies. High
emissions of haze pollutants (e.g., black carbon and organic matter) contribute greatly to the formation
of hazy weather in Beijing (Li and Han, 2016; Wu et al., 2016; Li et al., 2017). In addition, the
atmospheric and meteorological conditions as well as external and remote influences, such as Arctic
sea ice concentration (SIC), snow cover across Siberia, and sea surface temperature (SST), should also
be taken into consideration (An et al., 2019; Wang et al., 2020). Atmospheric circulations that are
favorable for hazy weather in Beijing include a weak East Asian winter monsoon (EAWM), a shallow



East Asian trough and a northward shift of the East Asian jet (Chen and Wang, 2015; Zou et al., 2017;
Pei et al., 2018; Wang et al., 2020). These circulations tend to reduce the intrusion of cold air to Beijing,
and hence result in poor ventilation conditions in winter (Zou et al., 2017; Pei et al., 2018). Furthermore,
teleconnection patterns and wave trains also have potential impacts on haze over Beijing (Yin and
Wang, 2017; Yin et al., 2017; Chen et al., 2019; Zhang et al., 2019; Chen et al., 2020a, 2020b; Lu et
al., 2020). Yin et al. (2017) analyzed the roles of the positive phases of the East Atlantic/West Russia
(EA/WR+) pattern, the western Pacific pattern, and the Eurasia pattern in the increased number of hazy
days over the North China Plain. They found that these climatic anomalies could lead to meteorological
conditions that are conducive to the formation of haze pollution through modulating the anticyclonic
anomalies over north China (Yin and Wang, 2017; Yin et al., 2017). The positive phase of the Arctic
Oscillation (AO+) pattern can also increase the number of hazy days in Beijing (Lu et al. 2020). Years
with high AO indices are accompanied by a weakened East Asian trough and a weakened Siberian
high, which suppress the vertical diffusion of haze pollutants (Lu et al. 2020). Chen et al. (2019)
stressed the role of the positive phase of the North Atlantic Oscillation (NAO+), which is related to
AO+, in inducing an anticyclone over northeast China, which favors haze pollution in north China in
spring. Chen et al (2020b) also looked at autumn haze, and found the relative importance of the external
drivers seems to differ across the individual months of September, October and November.
Additionally, the winter haze weather in Beijing is directly affected by the local meteorological
conditions. Many studies have suggested that static and relatively warm air, weakened northerly or
even southerly winds, decreased relative humidity, temperature inversions and downward air motion
in the planetary boundary layer can suppress the dispersal and advection of haze pollutants (Wang et
al., 2014; Zhang et al., 2014; Zhang et al., 2016; Wu et al., 2017; Wang et al. 2019; Zhong et al., 2019).





In line with the suggestion above a reduction in Arctic sea ice in autumn could increase
subsequent winter haze days through weakening wave activity over eastern China (Wang and Zhang,
2015; Zou et al., 2017). Zou et al. (2017) also revealed that increased Eurasian snowfall in earlier
winter leads to regional circulations unfavorable to the ventilation of pollutants. Furthermore, autumn
Beaufort Sea ice can be closely connected with the number of early-winter haze days in north China
(Yin et al., 2019a; Li and Yin, 2020), while an increase in early-winter Chukchi Sea ice can intensify
February haze pollution in north China (Yin et al., 2019b). The changes in sea ice in both the Beaufort
Sea and the Chukchi Sea are linked with hazy weather in north China via modulated large-scale
atmospheric circulations (e.g., the East Asian trough and teleconnections). SSTs in both the Atlantic
and Pacific have potential impacts on the occurrence of hazy weather (Xiao et al., 2015; Pei et al.,
2018; Wang et al., 2019). Winter haze days in China are also associated with SST anomalies over the
North Atlantic on decadal and interannual timescales via the Atlantic multidecadal oscillation, and are
also related to SSTs over the South Atlantic on interannual timescales by anomalous southerly airflow
(Xiao et al., 2015). Pei et al. (2018) found that positive SST anomalies over the northwest Pacific are
conducive to more winter haze days in Beijing by weakening the EAWM system. Wang et al. (2019)
also suggested that the interannual variability in autumn haze days in the Beijing–Tianjin–Hebei region
is associated with a wave train induced by SSTs in the North Atlantic subtropics and a local meridional
cell induced by SSTs in the western North Pacific.
Persistent haze events correspond to continuous pollution for several days, which not only has a
broad impact on human life (through traffic jams, for example), but also threatens human health to a
deeper degree. In this study, we gain a better understanding of the physical processes and mechanisms
of the persistent haze over Beijing. Local meteorological conditions are directly associated with hazy



weather and usually show diurnal variations (Zhang et al., 2014; Li et al., 2018b; Li et al., 2019), while
external forces, which vary slowly, play key roles in explaining the interannual/interdecadal
variabilities in hazy weather (Wang et al., 2020). On intra-seasonal timescales, large-scale atmospheric
circulations (e.g., teleconnections and wave trains) can bridge the timescales of local meteorological
conditions and external forces. Atmospheric circulations can be modulated by external forces and
meanwhile can influence local meteorological conditions. In previous studies, the atmospheric patterns
associated with hazy weather were mainly obtained from linear correlation or composite analysis based
on interannual or longer timescales (Yin and Wang, 2017; Yin et al., 2017; Chen et al., 2019; Lu et al.,
2020). However, the evolution of these atmospheric circulations and their roles in the formation of
hazy weather (especially persistent haze events) from daily to intra-seasonal timescales are not clear.
Thus, a more in-depth analysis of daily changes is needed to build a more solid relationship between
haze and atmospheric circulation, and to provide us with more hints for haze forecasting if certain
circulation patterns could be verified several days before persistent haze events. This will help to
improve predictive skills for persistent haze over Beijing. In this study, we focus mainly on persistent
winter haze over Beijing and the corresponding large-scale atmospheric circulations from the
perspectives of interannual and daily timescales. In particular, we are interested in the role of the
NAO+ pattern in the early stage of the formation of haze events. We also investigate the conditions of
SSTs and Arctic Sea ice that have been proposed as drivers of large-scale atmospheric circulations.
Furthermore, persistent haze events related to NAO+ and NAO− patterns are examined and compared.
**2 Data and methods**
**2.1 Data**
The observed relative humidity and visibility at 20 stations in Beijing at four local times (02:00,





08:00, 14:00 and 20:00) each day from 1980 to 2016 during the winter (December, January and
February (DJF)) are obtained from quality-controlled station observations at the National Mereological
Information Center of China. Daily means of these variables are then calculated. Reanalysis data are
taken from the European Centre for Medium-Range Weather Forecasts (ECMWF) ERA-Interim on a
$1\degree \times 1\degree$ grid (Dee et al., 2011). The variables include daily geopotential height at 500 hPa (Z500),
horizontal winds (at 500, 850 and 925 hPa), temperature (at 500, 700, 850 and 925 hPa), relative
humidity (at 500, 700, 850 and 925 hPa), SST, sea level pressure (SLP) and boundary layer height
(BLH). We also used monthly SST and SIC data from the Hadley Centre Global Sea Ice and Sea
Surface Temperature (HadISST) dataset (Rayner et al., 2003).

We use the normalized NAO index and EA/WR index from the National Oceanic and

Atmospheric Administration/Climate Prediction Center (NOAA/CPC). Our investigation also employs
a second NAO index (referred to hereafter as a modified LW03 NAO index), which is a modification
of that proposed by Li and Wang (2003). Those authors defined an NAO index as the difference in the
normalized SLP, zonally averaged from 80° W to 30° E, between 35° N and 65° N. They commented
that this measure 'provides a much more faithful and optimal representation of the spatial–temporal
variability associated with the NAO'. We have made this modification for our study because this
longitudinal range includes both the North Atlantic and part of the European continent. Atmospheric
circulations in our investigation involve both a blocking anticyclone over Europe (EB) and an NAO
pattern. When the anticyclone over Europe and the NAO pattern occur concurrently, the NAO index
of Li and Wang (2003) may have difficulties in interpreting the real NAO pattern, since the surface
pressure centered in the north of the NAO pattern is easily distorted by the large geopotential height
anomalies over Europe. To avoid this complexity, we have slightly modified the Li and Wang (2003)



definition by conducting our sector averaging over 80° W–10° W, instead of the original 80° W–30°
E. Such a modification is supported by detailed NAO studies (Yao and Luo 2014; Luo et al. 2014; Yao
et al. 2016), which reveal the disadvantage of the prevailing NAO indices in the identification of the
spatial structures of the NAO pattern. They also suggested that the zonal position and inclination of
the NAO dipole could change the occurrence of extreme weather events.
**2.2 Definition of persistent haze events**
Haze is generally defined in terms of relative humidity and visibility (World Meteorological
Organization, China Meteorological Administration, UK Met Office). In the "Specifications for
surface meteorological observations" compiled by the China Meteorological Administration (2004),
haze is defined as a weather phenomenon with a large amount of extremely fine dust particles evenly
floating in the air, which reduces the horizontal visibility to less than 10 km. Since visibility can also
be reduced by fog, a relative humidity threshold is applied to distinguish hazy weather from fog. Many
previous studies (Wu, 2006, 2008; Yang et al., 2016; Pei et al., 2018; He et al., 2018) proposed that
fogs are associated with a relative humidity greater than 90%. Therefore, we use a daily mean relative
humidity of less than 90% and a visibility of less than 10 km as criteria to define winter haze days in
this study (Pei et al., 2018; Chang et al., 2020). Persistent haze events are defined as periods for which
haze occurs for at least 5 consecutive days.
**2.3 Atmospheric thermal stability index**
The atmospheric thermal stability index ( $A_I$ ) is defined according to Zhang et al. (2007), as:
$$A_I = (T_{850} - T_{500}) - [(T_{850} - T_{d850}) + (T_{700} - T_{d700}) + (T_{500} - T_{d500})]$$
where $T$ and $T_d$ represent the temperature and dewpoint, respectively, and the subscripts denote the
pressure level.





**3 Associated large-scale atmospheric circulations**

**3.1 Interannual variability**

To present the context of the relationship between Beijing winter haze weather and atmospheric circulation, in Fig. 1a we show the time series of the number of winter haze days (blue line) and detrended winter haze days (red line) from 1980 to 2016. The linear trend (+1.93 days/decade) of the number of winter haze days in Beijing is not significantly different from zero, which is consistent with previous studies (Chen and Wang, 2015; Pei et al., 2018, 2020), and its interannual variability is quite large. Many factors could affect the variation in number of winter haze days, including changes in emissions, changes in emission reduction measures and climate variables (e.g., meteorological conditions, atmospheric circulation and SSTs) (Dang and Liao, 2019; Wang et al., 2020; Pei et al. 2020). Pei et al. (2020) pointed out that anthropogenic emissions showed an increasing trend before 2012 and a decreasing trend after 2012, which can be attributed to a Clean Air Action Plan introduced in 2013, and the inconsistent trends could explain the absence of a significant trend in the number of hazy days. Thus, we undertake our investigation with detrended data to explore the influences which are associated with the interannual variation in number of hazy days. For the sake of reliability, long-term trends are removed for winter haze days and other variables in the following analyses.

We focus first on the role of atmospheric circulation in hazy weather from interannual timescales. The detrended winter haze days are significantly correlated with circulation patterns in the Z500 anomaly field (Fig. 1b). It is shown that a wave train with a wavenumber-3 structure dominates the mid-high latitudes. Three cyclones are situated over the Greenland region, the Ural region and the Sea of Okhotsk with positive geopotential height anomalies in between (Fig. 1b). Atmospheric teleconnections, namely the NAO+ pattern and the EA/WR+ pattern have been suggested to be



connected with winter haze days in Beijing (Yin and Wang, 2017; Yin et al., 2017; Chen et al., 2019).
Barnston and Livezey (1987) show the NAO+ pattern to be made up of negative geopotential height
anomalies in the high latitudes of the North Atlantic (Greenland) and positive anomalies over the
central North Atlantic, extending into the eastern United States and western Europe. The EA/WR+
pattern is associated with positive geopotential height anomalies over Europe and northern China, and
negative anomalies located over the central North Atlantic and the north of the Caspian Sea. Following
Barnston and Livezey (1987), we can say that Fig. 1b demonstrates an NAO+ pattern over the North
Atlantic, although this pattern is situated further to the west and co-occurs with a blocking anticyclone
over Europe. In addition, although a quadrupole mode from the North Atlantic to northern China shares
some similarities with an EA/WR+ pattern, there are also some differences. On the one hand, two
negative geopotential height anomalies are located over the northern North Atlantic and the northern
Ural region, which are further north than the two cyclones of the EA/WR+ pattern. On the other hand,
the EA/WR+ pattern shows a zonal wave train structure; and only when the cyclone (north center of
the NAO+ pattern) over the North Atlantic is excluded, can a tripolar mode from Europe to northern
China constitute a zonal wave train structure.

To further understand the relationships between winter haze days in Beijing and teleconnection

patterns of NAO and EA/WR, we present the annual variations in winter haze days, the NAO index
and EA/WR index from NOAA/CPC, and the modified LW03 NAO index in Fig. 2. Their correlation
coefficients are calculated over the time periods 1980–2016, 1980–1999 and 2000–2016. Correlation
coefficients between winter haze days and the NOAA/CPC NAO index are 0.27, 0.16 and 0.33,
respectively, and only the correlation for the entire 1980–2016 period is significant at the 90%
confidence level (Fig. 2a). Considering that the NAO+ pattern assumes a more westerly position





because of the presence of the EB in Fig. 1b, this NAO index cannot fully reflect the real relationship
between the north–south dipole mode over the North Atlantic and winter haze days in Beijing. For the
reasons discussed earlier, our modified LW03 NAO index presents a clearer metric of the NAO for our
study. Correlation coefficients between winter haze days in Beijing and the modified LW03 NAO index
are considerably larger for the 1980–2016 and 2000–2016 periods, being 0.42 ($p < 0.01$) and 0.61 ($p$
$< 0.01$), respectively, while the correlation coefficient (0.08) is smaller (and nonsignificant) during the
period 1980–1999 (Fig. 2b). As for the correlation with the EA/WR index, it is slightly smaller than
that with the modified LW03 NAO index for the entire period (0.36, $p < 0.05$), while it is considerably
higher (0.57, $p < 0.01$) when only the first 20 years of the record are considered. Overall, we see that
the NAO+ pattern is more closely related to winter haze days in Beijing. However, it should be noted
that the EA/WR+ pattern has a closer relationship with hazy weather in Beijing before 2000, while the
NAO+ pattern is dominant after that date. It is also worth noting that the extreme numbers of winters
haze days more closely correspond with the magnitude of the NAO index. Figure 2 shows that the
extremely low numbers of hazy days in the winters of 1995, 2003 and 2010 occur simultaneously with
small NAO values of −1.11 standard deviation (std dev), −1.11 std dev and −1.61 std dev, respectively,
while the winter of 2013 with an extremely high number of hazy days has a large NAO value of 2.42
std dev. However, such correspondences could not be found in the values of the EA/WR index. Thus,
we propose that the modified LW03 NAO index is better correlated with winter haze days in Beijing,
especially in recent decades (2000–2016), and better explains the extreme values of winter haze days
in Beijing.
**3.2 Large-scale and local daily atmospheric structure before and after persistent haze events**

To cast further light on the above results, we have identified all the persistent haze events over





the period 1980–2016, and explored the daily progression of the circulation structures which lead up
to (out to 10 days) and follow (out to 8 days) the 65 identified persistent haze episodes. To accomplish
this, we formed composites (across the 65 episodes) of daily Z500 anomalies and the horizontal
components of wave activity flux (Takaya and Nakamura, 2001). The sequence of the composites is
shown in Fig. 3, where Day 0 denotes the day with the minimum visibility within a persistent haze
event. From Day −10 to Day −8, an NAO+ pattern can be identified over the North Atlantic, and wave
activity flux propagates downstream from the north pole of the NAO+ pattern to a weak anticyclone
over Europe. Compared with the NAO pattern in Li and Wang's work (2003), the north–south dipole
mode of the NAO pattern here is situated more to the west and is accompanied by an anticyclone over
Europe. This is why we use a modified NAO index in this study, which was introduced in Sect. 2.
From Day −6 to Day −4, the weak anticyclone over Europe gets stronger and a cyclone develops over
western Russia following the propagation of wave activity. Simultaneously, the NAO+ pattern
weakens and shows a northeast–southwest inclination. From Day −3 to Day −2, an EA/WR+ pattern
is becoming obvious at the mid to high latitudes. At this time, a zonal wave train that propagates from
a cyclone over the North Atlantic, through an anticyclone over Europe and a cyclone over the west of
Lake Baikal, to East Asia leads to the formation of an anticyclone over northeastern China. On Day
−1, an anticyclone forms over the Gulf Stream region, which continues to provide wave activity to the
development of downstream circulation. This anticyclone sustains the wave train propagating to
northeastern China. As a result, the anticyclone over northeastern China persists for a total of 7 days
from Day −3 to Day 3 and then moves eastward to the North Pacific. The wave train structure dissipates
from Day 4, and the anomalous atmospheric circulation around Beijing becomes weaker.

From the daily evolution of atmospheric circulation, we find that certain teleconnection patterns





(NAO+ and EA/WR+) and wave trains lead to persistent haze events in Beijing, since the formation
of teleconnection patterns is closely related to zonal winds, and zonal winds act as waveguides for the
propagation of wave trains (Ambrizzi et al., 1995; Fang et al., 2001; Athanasiadis et al., 2010; Wang
and Zhang 2015; Martinez-Asensio et al., 2016; Wirth et al., 2018). Zonal wind anomalies at 300 hPa
are composited between Day −10 and Day 5 for the 65 persistent haze events to explain the evolution
of large-scale circulations and to ascertain whether potential predictors exist (Fig. 4a). Corresponding
to the propagation of an NAO+ pattern and an EA/WR+ pattern during the formation of persistent haze
events in Beijing, significantly intensified or weakened anomalous zonal wind centers could be clearly
identified from the North Atlantic to East Asia. The negative–positive–negative zonal wind tripolar
mode over the North Atlantic reflects the NAO+ pattern, while the arched structure of the anomalous
zonal wind from Greenland, passing through high latitudes to northern China, manifests in the
EA/WR+ pattern. To track the daily variations in anomalous zonal winds, significantly increased zonal
wind anomalies over the central North Atlantic, the Scandinavian Peninsula and the north of China are
taken as three indicators (Fig. 4b). The amplitude of the anomalous zonal wind over the central North
Atlantic peaks at Day −8, which corresponds to the prevalence of the NAO+ pattern from Day −8 to
Day −10 in Fig. 3. Although the zonal wind in this region weakens after Day −8, it retains significantly
high values until Day 4. Simultaneously with the significantly increased zonal wind over the central
North Atlantic, the difference in the zonal wind over the south of the Scandinavian Peninsula and the
zonal wind over north China also show significant positive anomalies starting from Day −11 and Day
−5, respectively. The difference in the zonal wind over the south of the Scandinavian Peninsula reaches
its highest point on Day −2 when the EA/WR+ pattern can be clearly identified in Fig. 3, while the
anomalous zonal wind over the north of China reaches a peak on Day −1. From the analyses above,





we can see that the three indicators of anomalous zonal winds could well reflect the sequential
occurrence of teleconnection patterns related to persistent haze events. These indicators also express
the downstream propagation of a wave train from the North Atlantic to northern China. Since the
intensified zonal winds over the central North Atlantic, the Scandinavian Peninsula and the north of
China are prior to the persistent haze events, they have potential value as predictors for persistent haze
events.
From the Day 0 panel of Fig. 3 we note that Beijing is under the influence of an anticyclone,
whose center is a short distance to the northeast of the city. This anticyclone could influence the
accumulation of haze pollutants in Beijing by modulating the meteorological conditions (Zhong et al.,
2019). We now turn our attention to the spatial–temporal variations in the local meteorological
conditions related to the anticyclonic circulation (Fig. 5). Figure 5a depicts the spatial distribution of
composite anomalous 850 hPa horizontal wind velocity and SLP for the 65 persistent haze events.
Beijing is located at the border of anomalous high-pressure areas over mainland China and anomalous
low-pressure areas situated over the northwestern Pacific. The weakened pressure gradients between
mainland China and the coastal regions indicate a weakened EAWM (Pei et al. 2018). Beijing is
controlled by anomalous southerly winds and the wind speeds are significantly less than normal.
Together with the vertical downdraft of the anticyclone at 500 hPa, these meteorological conditions
militate against the ventilation of pollutants. In addition, daily variations in composite meridional wind
component (V), wind velocity, relative humidity, temperature at 925 hPa, BLH and $A_I$ in Beijing are
consistent with the poor dispersal of pollutants (Fig. 5b). The BLH decreases rapidly after Day −5,
reaches a minimum at Day 0 and then rapidly increases, whereas the temperature shows the opposite
variations. The relative humidity and $A_I$ reach their largest values on Day 1. The smaller BLH and





$A_l$ correspond to stable weather conditions, which also inhibit the dispersal of pollutants. Furthermore,
the increased temperature and relative humidity are conducive to the hydroscopic growth of pollutants,
which could further aggravate the hazy weather. Though these local meteorological elements are most
directly related to persistent haze events in Beijing, they do not show prominent prior changes.
**4 External influence of sea surface temperature and sea ice concentration**
As noted earlier, previous studies showed that atmospheric circulations are affected by external
influences, such as SST and Arctic SIC (Wang and Zhang, 2015; Yin et al., 2017; Chen et al., 2019).
Since many of these external forces vary slowly and have significant memory, they may potentially
present precursors to persistent haze events. In this section, we relate SST in the North Atlantic and
SIC in the Greenland Sea and Barents–Kara Sea with persistent haze events in Beijing via atmospheric
circulations.
Below we refer to the number of persistent haze days in a winter as the sum of hazy days
belonging to persistent haze events of the winter. This measure reflects the number of persistent haze
events as well as their duration. The composite distributions of SST anomalies and SIC anomalies of
11 winters (1988, 1996, 1998, 2005, 2007, 2008, 2011–15) with more than 15 persistent haze days and
11 winters (1980, 1982–84, 1986, 1987, 1994, 1995, 1999, 2003, 2010) with no more than 5 persistent
haze days, and their differences are shown in Fig. 6.
As shown in Fig. 6a, the SST anomalies associated with the most persistent haze days show a
north–south tripolar mode in the North Atlantic, with positive anomalies in the Gulf Stream region and
negative anomalies at the southwest of the Canary Islands and regions around Greenland, which is
consistent with the NAO+ pattern. Simmonds and Govekar (2014) noted that warm SST in the Gulf
Stream region gives rise to a teleconnection pattern with a node in the Barents Sea and another over



Eurasia. Significant SST anomalies can also be seen in the northwestern Pacific, with the negative SST
anomalies from Yellow Sea to the Sea of Japan and positive SST anomalies to the east of the
Philippines. These SST anomalies are difficult to explain the atmospheric anticyclone from mainland
China to the northwestern Pacific, which is directly related with persistent haze events in Beijing. For
SST anomalies of winters with fewer persistent haze days, a north–south tripolar mode in the North
Atlantic could be found opposite to that in Fig. 6a, though the anomalies are not significant (Fig. 6b).
In addition, significant positive SST anomalies are located at the Greenland Sea and the Barents Sea
(Fig. 6b). The structure of the SST anomalies in the northwest Pacific is similar to that in Fig. 6a, but
with much smaller magnitude. Figure 6c, which shows the difference between these two anomalies
plots, further highlights the distribution of SST anomalies that are associated with more persistent haze
days: a prominent north–south tripolar mode over the North Atlantic and negative SST anomalies over
the Greenland Sea and Barents Sea.

SIC anomalies in the Greenland Sea and Barents–Kara Sea also show significant differences

between winters with the greatest number of persistent haze days and winters with the lowest number
of persistent haze days (Fig. 6d–f). The Greenland Sea and the southern part of the Barents Sea are
dominated by positive SIC anomalies and the Kara Sea shows negative SIC anomalies, though they
are not significant (Fig. 6d). The only positive SIC anomalies that differ significantly from zero are
found in the north of the Barents–Kara Sea and the southeast of Greenland (Fig. 6d). By contrast, for
the lowest number of hazy days composite significant negative SIC anomalies are located in the
Greenland Sea and the south part of the Barents Sea (Fig. 6e). The difference between Figures 6d and
6e also suggests that the decrease in sea ice in the Greenland Sea and the southern Barents Sea is not
conducive to a greater number of persistent haze days or haze events (Fig. 6f).





From the composite analyses above, we can see that SST anomalies in the North Atlantic and SIC
anomalies in the Greenland Sea and Barents–Kara Sea are associated with the occurrence of persistent
haze days or haze events. To examine the potential causalities here we analyzed the composite SST
anomalies and SIC anomalies averaged the period of 10 days prior to the first day of the 65 haze events.
Prior to the persistent haze events, SST anomalies exhibit a north–south tripolar mode in the North
Atlantic (Fig. 7a). This pattern differs from that shown in Fig. 6a; the tripolar mode is located further
south, and has significant positive anomalies around the Florida peninsula and even into the Caribbean
Sea, and significant negative SST anomalies off the northeast coast of the US (Fig. 7a). Figure 7a also
shows significant negative SST anomalies in the Greenland Sea, and the Barents–Kara Sea. Turning
to the SIC composite anomalies for these 65 events, we observe significant increases in the Greenland
Sea and the Barents–Kara Sea (Fig. 7b). These prior distributions of SST and SIC anomalies are
conducive to setting up the atmospheric circulations and zonal winds that are favorable for the
formation of persistent Beijing haze events. The SST tripolar mode in the North Atlantic is favorable
for the formation of an NAO+ pattern, and the strongly increased zonal wind in association with the
NAO+ pattern encourages downstream wave train propagation. In concert with this, the decreased SST
and increased SIC in the Barents–Kara Sea region potentially enables a strengthening of the cyclone
upstream of the anticyclone over northern China, which is an essential part of the downstream wave
train. Furthermore, as in Fig. 6a, the significant negative SST anomalies in the northwestern Pacific
are not compatible with the anticyclone directly controlling persistent haze events in Beijing.
**5 Relationship between persistent haze events and the preceding phase of the NAO pattern**
The above analysis demonstrates that persistent haze weather in Beijing is closely related to the
NAO+ pattern, in that the regression pattern in Fig. 1b resembles an NAO+ pattern with a downstream



wave train more than an EA/WR+ pattern. Also, the NAO index has a stronger relationship with winter
haze days in Beijing than the EA/WR index, especially after 1999. More importantly, the NAO+
pattern leads the Day 0 of persistent haze events by 8–10 days, and corresponding SST signals could
be identified in the North Atlantic. Thus, the NAO+ pattern and related increased zonal winds and SST
anomalies are essential to the formation of persistent haze events in Beijing. We also note that SST
anomalies show opposite tripolar modes in Fig. 6a and Fig. 6b, meaning that the opposite SST mode
is unfavorable to the occurrence of persistent haze events in Beijing. This structure could force an
NAO− pattern through the air–sea interaction (Peng et al., 2002, 2003; Frankignoul and Kestenare,
2005; Luo et al., 2016; Jing et al., 2019).

In light of the above, we comment that, even though regression and composite analyses (e.g., Fig.

1) might suggest a strong link between haze events and the NAO+, it is important to remember that
such analyses identify *necessary* but not *sufficient* conditions (see Boschat et al., 2016). This leads us
to the question as to whether there are persistent haze events related to the NAO− pattern and what the
differences are between these haze events and those related to the NAO+ pattern. First, the daily
evolutions of the NAO index for the 65 persistent haze events (black line), 43 NAO+-related persistent
haze events (red line) and 22 NAO−-related persistent haze events (green line) are shown in Fig. 8.
The composite NAO index for all haze events reaches its highest value on Day −8, which is consistent
with Fig. 3 where the NAO+ pattern is more prominent from Day −10 to Day −8. Thus, we select the
NAO+-related (NAO−-related) persistent haze event if its averaged NAO index during the period of
Day −10 to Day −6 is above (below) zero. As shown in Fig. 8, the daily evolution of the composite
NAO index for NAO+-related and NAO−-related persistent haze events generally shows the opposite
character. However, the peak value (0.90) of the NAO index for NAO+-related persistent haze events




also appears on Day −8, whose absolute value is larger than that of the valley value (−0.68) of the
NAO index for NAO−-related persistent haze events on Day −10. Corresponding SST and SIC
anomalies of the NAO+-related case and the NAO−-related case are also compared in Figures 8b–8d.
We use a difference in the SST anomalies between the two regions A (40° N–55° N, 70° W–20° W)
and B (22° N–37° N, 85° W–35° W) marked in Fig. 7a to signify the SST tripolar pattern in the
North Atlantic, which could be related to an NAO+ pattern. Daily evolutions of the SST anomaly
difference show significant increases starting from Day −11 and Day −9 for the NAO+-related case
and all persistent haze events, respectively, whereas the difference in SST anomalies for the NAO−-
related case shows only a weak fluctuation (Fig. 8b). As for the SIC anomaly in the Barents–Kara Sea,
it has a significant maximum amplitude leading the Day 0 of NAO+-related persistent haze events by
12 days (Fig. 8d), while the SIC anomaly in the Greenland Sea shows significant high values during
the period from Day −15 to Day 15 of NAO−-related persistent haze events and its peak appears on
Day −3 (Fig. 8c). Thus, we can conclude that the SST anomalies in the North Atlantic and SIC
anomalies in the Barents–Kara Sea show preceding signals for NAO+-related persistent haze events,
and the NAO−-related persistent haze events are associated mainly with increased SIC anomalies in
the Greenland Sea.
Furthermore, a comparison between the evolution of atmospheric circulations for persistent haze
events related to the NAO+ pattern and the NAO− pattern is made in Fig. 9. For the NAO+-related
case, the development of atmospheric circulation is similar to that in Fig. 3, while the NAO+ pattern
is more prominent (Fig. 9a). Moreover, the persistent NAO+ pattern over the North Atlantic
corresponds to the significantly strengthened dipole mode in the SST, as suggested in Fig. 8b, which
sustains the propagation of the wave train; and the increased SIC (increased SST) in the Barents–Kara





Sea could affect the cyclone over Siberia, a node of the wave train. In Fig. 9b, the NAO− pattern is
situated over the North Atlantic on Day −6, whose north anticyclone corresponds to the increased SIC
anomalies at the Greenland Sea. The wave activity originates mainly from the north center of the
NAO− pattern (Day −6). As the wave activity flux propagates downstream, an anticyclone forms over
northeastern China on Day −2 and reaches its greatest intensity on Day 0. Compared with the evolution
of the wave train in Fig. 9a, that in Fig. 9b has a longer wavelength, but it is less organized and less
persistent. This suggests that the preceding NAO+ pattern could increase the occurrence of persistent
haze events compared to the NAO− pattern.

To confirm the role of the NAO+ pattern in the occurrence of persistent haze weather in Beijing,

we also compare the number of persistent haze days and persistent haze events that occur in NAO+
winters and NAO− winters. The NAO+ winters (NAO− winters) are classified with four sets of criteria
to make our results more solid and convincing. If the normalized NAO index of DJF is above the
50th/75th/90th/95th percentile (below the 50th/25th/10th/5th percentile), NAO+ winters (NAO−
winters) are selected. It is found that the number of persistent haze days and persistent haze events
happening in NAO+ winters are much larger than in NAO− winters (Fig. 10). The discrepancy is
especially large for the 90th (10th) and 75th (25th) percentiles. For these two sets of criteria, the
number of hazy days in NAO+ winters is more than twice that in NAO− winters, while the number of
haze events in NAO+ winters is nearly double that in NAO− winters. This also suggests that the haze
events in NAO+ winters have longer durations. Though the differences in the number of persistent
haze days (events) for the 95th (5th) and 50th (50th) percentiles, which contain the lowest number of
haze days (events) and the greatest number of haze days (events), are not as apparent as the other two
sets of criteria, NAO+ winters still have a greater number of persistent haze days (events). This





confirms our finding that the NAO+ pattern is more conducive to the occurrence of persistent haze
weather in Beijing.
**6 Conclusions and discussions**

In this study, we have focused mainly on persistent winter haze events in Beijing from the

perspective of large-scale atmospheric circulation (e.g., teleconnections and wave train). We first
presented the atmospheric circulation correlated with winter haze days in Beijing on interannual
timescales. It is consistent with previous studies that an NAO+ pattern and an EA/WR+ pattern are
related to the frequency of winter haze (Yin et al. 2017; Yin and Wang, 2017; Chen et al. 2019).
Furthermore, our new result shows that the NAO+ pattern has a stronger relationship with winter haze
days, especially after 1999. In addition, the NAO+ index corresponds better to the winters with the
greatest and lowest number of hazy days.

To increase our understanding of the relationship between winter haze weather in Beijing and

atmospheric circulation, we investigated the daily evolution of the composite circulation patterns and
wave activities for 65 persistent haze events. A transmission of atmospheric circulation demonstrated
a wave train propagating from the North Atlantic to northeast China. The NAO+ pattern performs as
the origin of this transmission from Day −10 to Day −8, followed by an EA/WR+ pattern from Day
−3 to Day −2. We also highlighted the daily variations in zonal winds linked to atmospheric circulation.
It was revealed that the zonal wind over the central North Atlantic, which is coupled with the NAO+
pattern, reached its largest value on Day −9, and the zonal wind downstream over the Scandinavian
Peninsula and the north of China showed significantly increased values after Day −5, indicating the
propagation of the wave train. Thus, the NAO+ pattern and the downstream wave train, as well as the
successive increases in zonal winds could serve as potential predictors for persistent winter haze events





in Beijing.
It should be noted that the last node of the wave train is an anticyclone over northeastern China,
which is directly associated with the formation of haze over Beijing (Zhong et al., 2019). We then
investigated the local meteorological conditions associated with the anticyclone that are conducive to
the accumulation of haze pollutants over Beijing. The negative anomalous pressure over mainland
China and the positive anomalous pressure over the coastal regions suggest a weakened EAWM.
Correspondingly, there are southerly winds and decreased wind speed in Beijing. Furthermore, the
BLH decreases, while $A_I$ increases during the haze episodes. These conditions can suppress the
ventilation and dispersal of haze pollutants. In addition, the anomalously high values of relative
humidity and temperature at 925 hPa could worsen the hazy weather by accelerating the hydroscopic
growth of pollutants. However, these meteorological conditions show almost simultaneous variations
with the process of persistent haze events; thus, it is difficult to use them as predictors.
Atmospheric circulations can be forced by external influences and these, in turn, may provide
precursors to haze events. Analyses revealed that the SST anomalies in the North Atlantic and SIC
anomalies in the Greenland Sea and Barents–Kara Sea are related to winter haze weather in Beijing on
both interannual and intra-seasonal timescales. On interannual timescales, winters with a greater
number of persistent haze days correspond to a north–south tripolar mode over the North Atlantic. On
intra-seasonal timescales, a more southerly SST tripolar mode is also prominent over the North Atlantic,
and the Greenland Sea and Barents–Kara Sea also present significant SST and SIC anomalies 10 days
prior to the persistent haze events. For a more in-depth understanding, the relationship between the
preceding NAO pattern, SST and SIC anomalies and the persistent haze events was highlighted by a
comparison of NAO+-related haze events and NAO−-related events. We found that the NAO+-related





case corresponds to prior SST signals in the North Atlantic and increased SIC anomalies in the
Barents–Kara Sea, while the NAO−-related case had large SIC anomalies only in the Greenland Sea.
We proposed that these SST and SIC anomalies could also provide hints for the prediction of persistent
haze events in Beijing, especially for the NAO+-related case, considering that the SST tripolar mode
in the North Atlantic can affect the NAO+ pattern and the change in the SST and SIC at the Barents–
Kara Sea promotes the downstream propagation of the wave train.

It has also been revealed that the number of NAO+-related haze events is nearly double the

number of the NAO−-related cases, and the atmospheric circulation and the propagation of wave
activity are more organized for the NAO+-related case. These results indicate that persistent haze
events are more likely to occur with the NAO+ pattern. To verify this proposition, we further compared
the number of persistent haze days and persistent haze events in NAO+ winters and NAO− winters
with four sets of criteria. It was found that the numbers of both haze days and haze events are larger in
NAO+ winters, especially for the 95th (5th) and 50th (50th) percentiles.

SSTs in the Atlantic (Xiao et al. 2015; Wang et al. 2019), SICs in the Arctic (Wang and Zhang,

2015; Yin et al. 2019a, 2019b) as well as the large-scale atmospheric circulations (Yin and Wang 2017;
Yin et al. 2017; Chen et al. 2019) have been pointed out to be closely associated with the winter haze
in Beijing from decadal timescales down to intra-seasonal timescales. These studies well captured
certain modes of SSTs, SICs and atmospheric circulations that are significant during the haze episodes,
but did not reveal their lead-lag relationships on daily time scales. Through analyzing the day-by-day
evolutions, our investigation has revealed that the NAO+ pattern, part of an atmospheric transmission,
and the associated intensified zonal wind, SST mode and SIC anomalies show potential prediction
abilities for persistent haze events, which provides us with a possible approach for the prediction of



winter haze weather in Beijing. The preceding signals in the SST, SIC and atmospheric transmission
over the North Atlantic and Eurasia may serve as a predictor for persistent haze events. In further
studies, it will be necessary to qualitatively estimate the contributions of emissions and atmospheric
transmission patterns in haze formation. This could be useful for the government to take effective
measures to reduce emissions, to reduce the occurrence of the winter haze weather in Beijing. In
addition, it is also worthwhile figuring out the reasons why the NAO index (EA/WR index) has a
stronger correlation with winter haze weather in Beijing after 1999 (before 2000) from the perspective
of climate change.

***Data availability.*** Daily atmospheric and land-surface data were downloaded from the ECMWF ERA-
Interim data archive (http://www.ecmwf.int/en/research/climate-reanalysis/era-interim) (ERA-Interim,
2017). Monthly sea surface temperature data and sea ice concentration data are available from the Met
Office Hadley Centre datasets (https://www.metoffice.gov.uk/hadobs/hadisst/data/download.html)
(Met Office, 2017). The ground observations were taken from the National Meteorological Information
Center of China (http://data.cma.cn/) (CMA, 2017). The NAO and EA/WR indices are from the
NOAA's Climate Prediction Center (http://www.cpc.ncep.noaa.gov/data/teledoc/telecontents.shtml)
(CPC, 2017). The modified LW03 NAO index can be obtained from the authors.

***Competing interests.*** The authors declare that they have no conflict of interest.

***Author contributions.*** YY designed the study. ML created the figures. ML and YY wrote the
manuscript. IS, DL and LZ gave constructive comments. LP provided and analyzed the observational



station data.

***Acknowledgments.*** The authors acknowledge support from the National Natural Science Foundation
of China (Grants 41975068 and 41790473) and the National Key Research and Development Program
of China (Grant 2016YFA0601802). Simmonds was supported by Australian Research Council Grant
DP16010997.

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

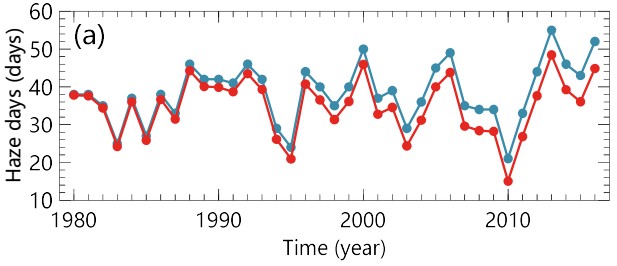


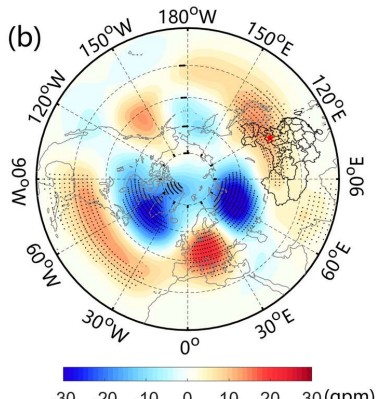


**Figure 1. (a)** Time series of winter haze days (blue) and detrended winter haze days (red) in Beijing

from 1980 to 2016. To aid visualization, a fixed value is added to all the detrended data so that the first

values of both series coincide. **(b)** Regressed Z500 anomalies (shading; units: gpm) against the

detrended winter haze days. The dotted areas show values that are above the 95% confidence level

based on the two-sided Student's *t*-test. For all plots in this study, the red star denotes the location of



Beijing.




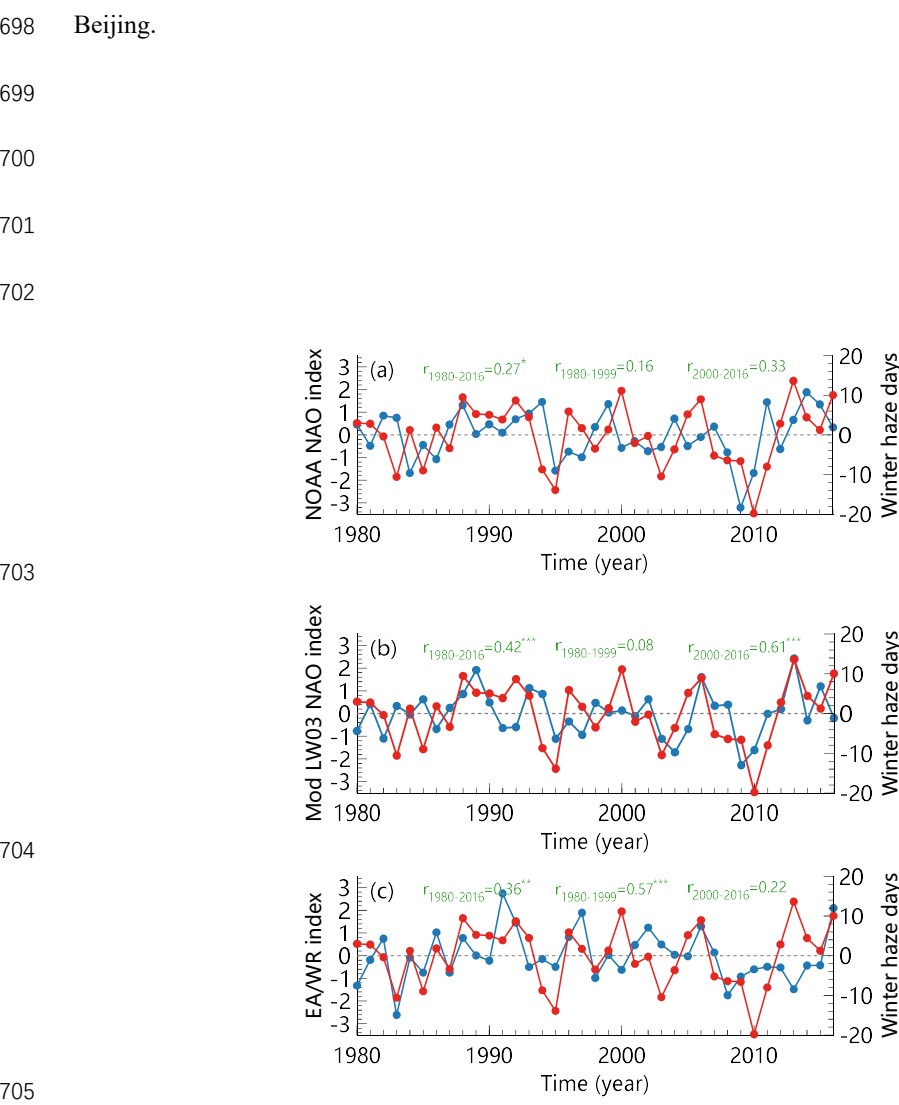




**Figure 2.** Time series of detrended winter haze days (red) and (a) NOAA NAO index (blue), (b)
modified LW03 NAO index (blue) and (c) NOAA EA/WR index (blue). Correlation coefficients
between winter haze days and the three indices during the periods 1980–2016, 1980–1999 and 2000–
2016 are labelled at the top of each panel. The 90%, 95% and 99% confidence levels for the Student's
*t*-test are denoted by one, two and three asterisks, respectively.




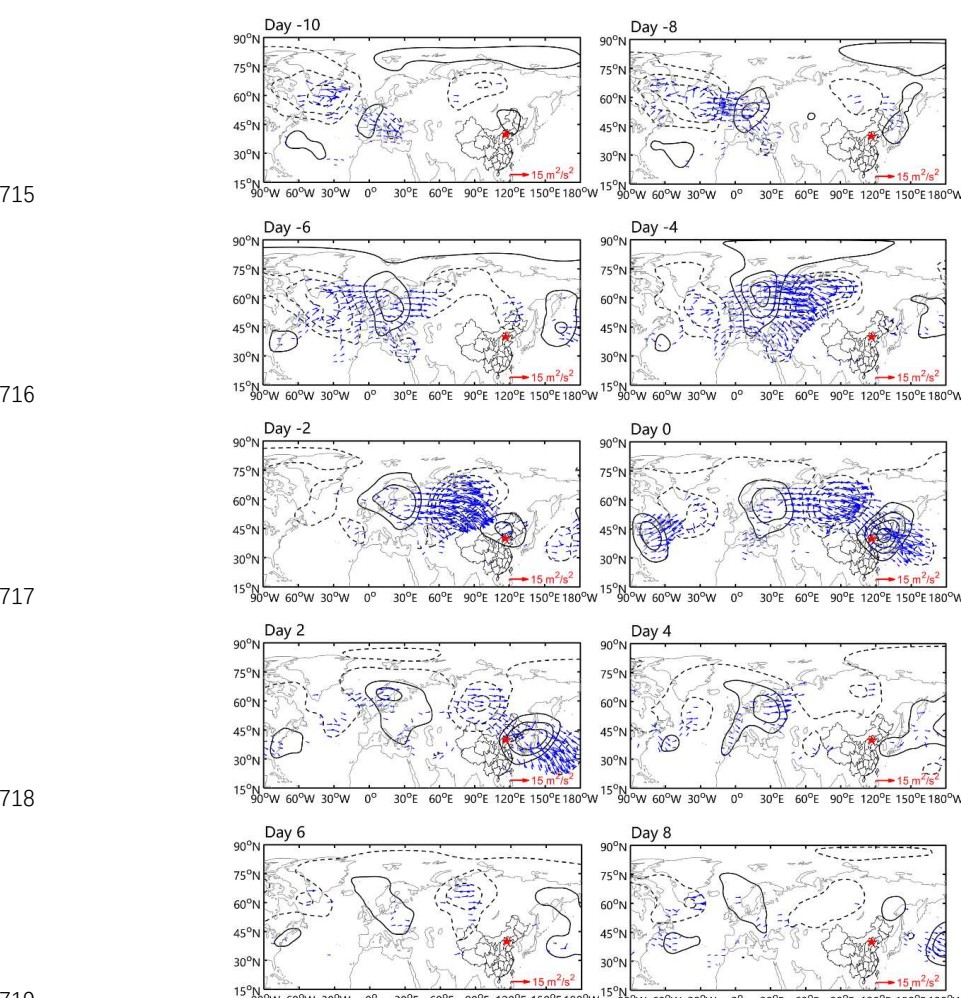

**Figure 3.** Instantaneous fields of the composite daily Z500 anomalies (contours, interval = 20; units:

gpm), and horizontal components of wave activity developed by Takaya and Nakamura (2001) (arrows;

units: m$^2$/s$^2$) from Day −10 to Day 8 for the 65 persistent haze events.









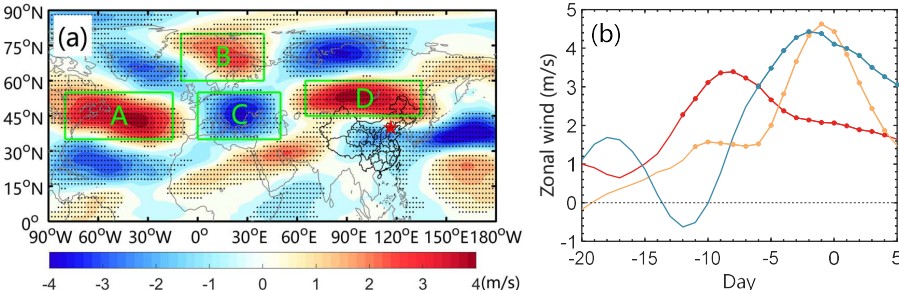



**Figure 4. (a)** Composite 300 hPa zonal wind anomalies (shading; units: m/s) between Day −10 and

Day 5 for the 65 persistent haze events. The dotted areas show values that are above the 95%

confidence level based on the two-sided Student's *t*-test. **(b)** Composite daily time series of region-

averaged 300 hPa zonal wind anomalies in region A (80° W–15° W, 35° N–55° N) (red) and region D

(65° E–135° E, 45° N–60° N) (orange) and the difference in region-averaged zonal wind anomalies

between region B (10° W–40° E, 60° N–80° N) and region C (0° W–50° E, 35° N–55° N) (blue). The

dots denote the days with values that are above the 95% confidence level for the two-sided Student's

*t*-test.













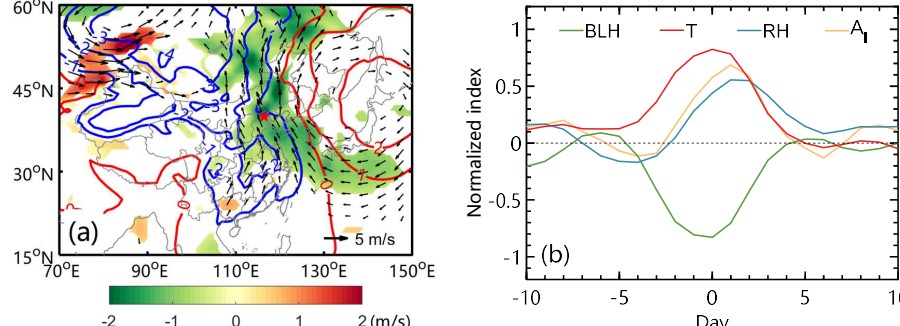


**Figure 5. (a)** Composite anomalous 850 hPa horizontal wind vector (vector), 850 hPa wind speed

(shading; units: m/s) and SLP (contour; units: hPa) between Day −2 and Day 2 for the 65 persistent

haze events. Only wind vectors above 1 m/s are plotted. **(b)** Composite daily time series of boundary

layer height (BLH, green), temperature (T, red), relative humidity (RH, blue) and $A_I$ (yellow) at 925

hPa averaged over the region of Beijing (115° E–118° E, 39° N–42° N, marked with a red star in **(a)**)

for the 65 persistent haze events.











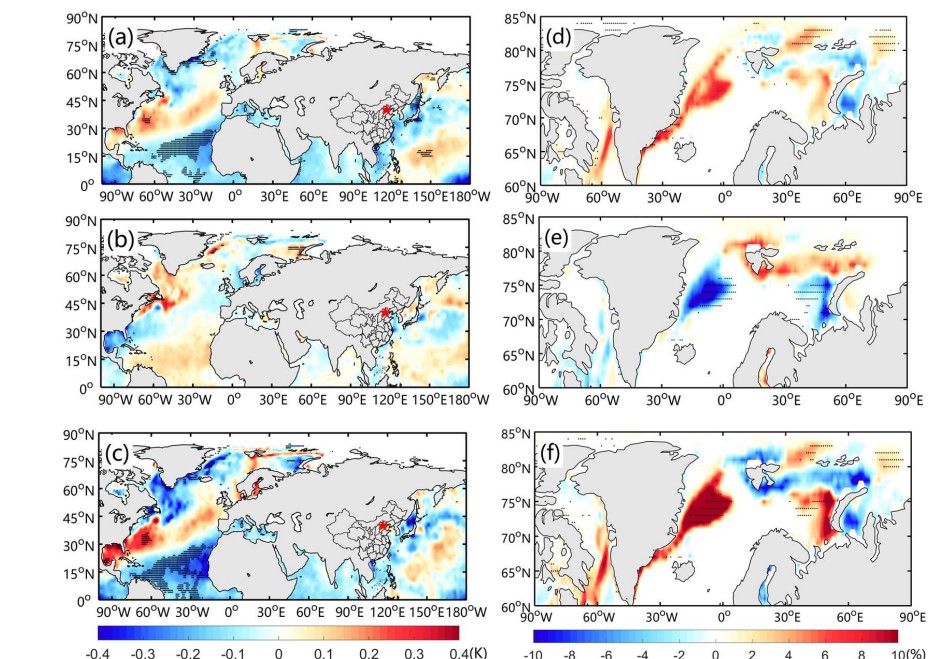




**Figure 6.** Composite SST anomalies and SIC anomalies for **(a, d)** winters (1988, 1996, 1998, 2005,

2007, 2008, 2011–15) with the greatest number of persistent haze days and **(b, e)** winters (1980, 1982–

84, 1986, 1987, 1994, 1995, 1999, 2003, 2010) with the lowest number of persistent haze days, and **(c,**

**f)** their differences. The dotted areas show values that are above the 95% confidence level based on a

Monte-Carlo test with 10,000 simulations.










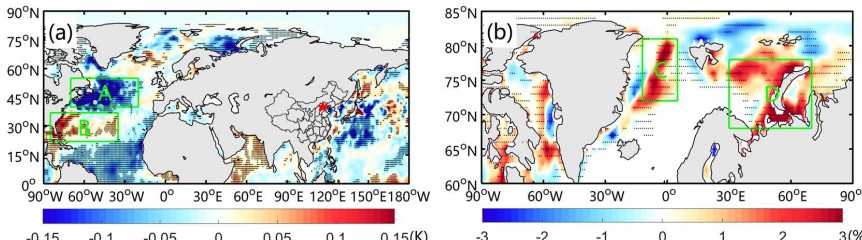

**Figure 7. (a)** Composite SST anomalies and **(b)** SIC anomalies for the 10 days before the first day of

the 65 persistent haze events. The dotted areas show values that are above the 95% confidence level

based on the two-sided Student's *t*-test. The green rectangles A and B in **(a)** denote a north region (40°

N–55° N, 70° W–20° W) and a south region (22° N–37° N, 85° W–35° W) over the North Atlantic,

and the green rectangles C and D in **(b)** denote the Greenland Sea (12° W–5° E, 72° N–81° N) and the

Barents–Kara Sea (30° E–70° E, 68° N–78° N), respectively





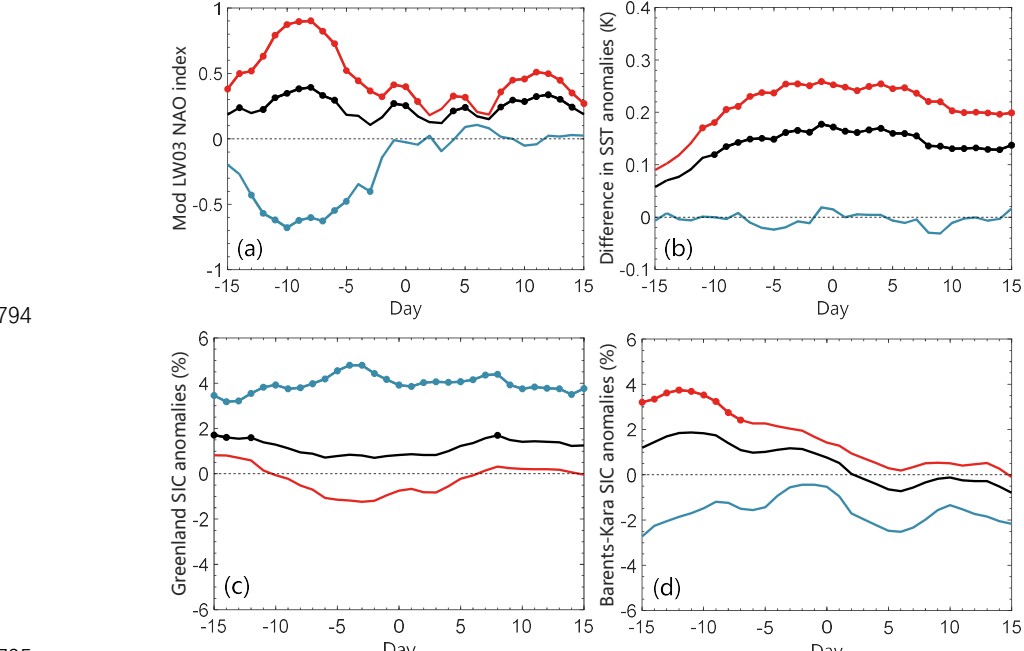

794

795

**Figure 8**. Composite daily time series of the **(a)** modified LW03 NAO index, **(b)** difference in the SST

anomalies over the two regions, as presented in Figure 7a, **(c, d)** region-averaged SIC anomalies in the

**(c)** Greenland Sea (marked by C in Figure 7b) and **(d)** Barents–Kara Sea (marked by D in Figure 7b)

for the 65 persistent haze events (black), 43 NAO+-related persistent haze events (red) and 22 NAO−-

related persistent haze events (blue) from Day −15 to Day 10. The difference in the SST anomalies is

calculated by subtracting the region-averaged SST anomalies in the north (rectangle A) from the

region-averaged SST anomalies in the south (rectangle B) (i.e., B minus A). The persistent haze event

that is related to NAO+ (NAO-) is selected when its averaged NAO index is positive (negative) from

Day -10 to Day -6 of the persistent haze event.

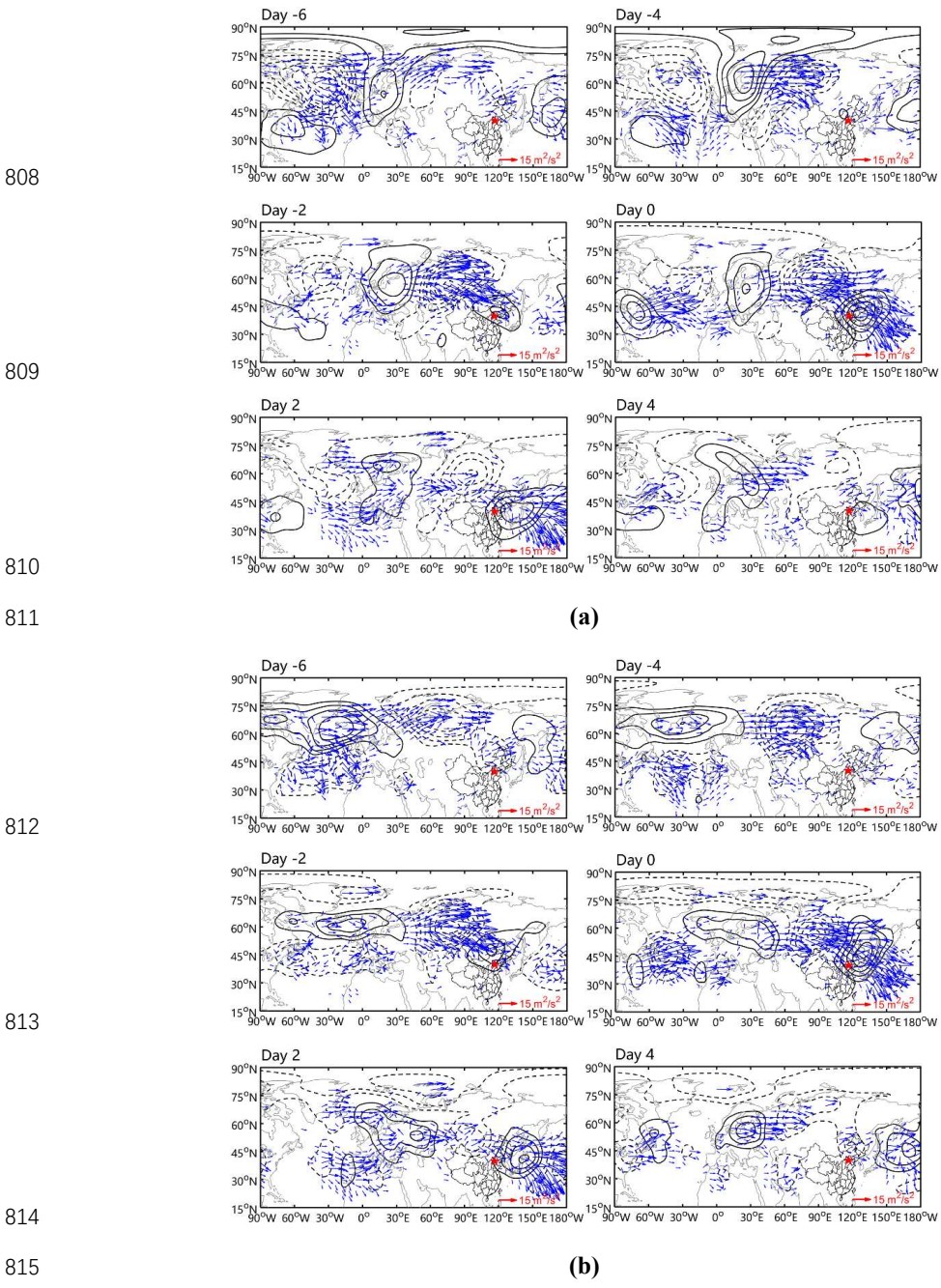





**(a)**





**(b)**

**Figure 9.** Instantaneous fields of composite daily Z500 anomalies (contours, interval = 20; units: gpm),
SAT anomalies (shading; units: K) and horizontal components of W (vectors; units: m$^2$/s$^2$) for





persistent haze events **(a)** related to NAO+ (43 events) and **(b)** related to NAO− (22 events) from Lag
−6 to Lag 4. Only W vectors above 2 m$^2$/s$^2$ are plotted.








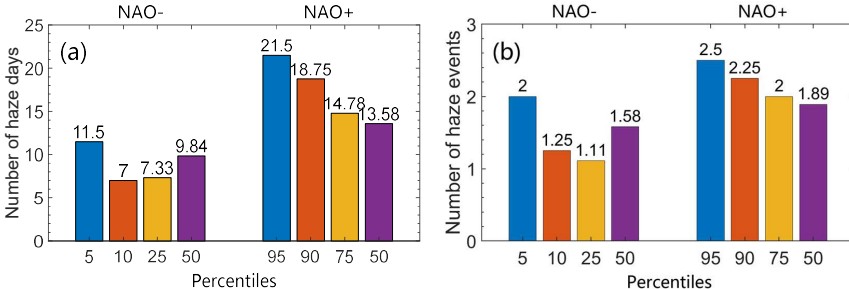


**Figure 10.** Averaged number of winter **(a)** haze days and **(b)** haze events during NAO+ winters and

NAO− winters. The blue, orange, yellow and violet bars on the left and right denote winters with an

NAO index larger than the 5th, 10th, 25th and 50th percentiles (NAO− winters) and winters with the

NAO index smaller than the 95th, 90th, 75th and 50th percentiles (NAO+ winters), respectively.

829