# Peer review of "Atmospheric transmission patterns which promote persistent"

_Atmospheric Chemistry and Physics, 2020_

## Author Comment (AC1)

Comment on acp-2020-823:

The manuscript titled "Atmospheric transmission patterns which promote persistent winter haze over Beijing" attempted to illustrate the relationship between North Atlantic oscillation and persistent haze in winter in Beijing, especially on the intraseasonal time-scale. Although the topic is interesting, many major revision must be revised.

**Response:** *Thanks for your comments. We have revised the manuscript according to your remarks, and our point-by-point responses to the comments are listed in detail below.*

Major comments:
Title: Possibly, the title is not accurate. From your title, I cannot read any valuable information about the NAO pattern that you focused on. I strongly suggest you to rephrase your title of this manuscript.

**Response:** *You make a good point here. Yes, this paper highlights the important role of NAO on atmospheric transmission patterns, but this is not reflected in the title. Therefore, we have revised the title of this paper in the revised version to better reflect the focus of this study. The title is now 'Linkages between the atmospheric transmission originating from the North Atlantic Oscillation and persistent winter haze over Beijing'.*

Line 17: "This study focuses mainly on the role of the NAO+ pattern, because the NAO index shows a closer relationship with winter haze frequency, especially after 1999". However, I did not find any physical explanations about this arguments that you emphasized in the Abstract and that seemed to be the reasons why you did this research.

**Response:** *We appreciate the point you are making here. We have removed this sentence in the Abstract, since we did not explain this relationship. Instead, we now present one of the rationales for this research by adding this "This study mainly focuses on the role of the WNAO pattern, because the WNAO+ pattern acts as the origin of the atmospheric transmission, 8–10 days prior to the persistent haze events."*

Line 18–20: The intraseasonal relationship between North Atlantic oscillation and persistent winter haze in Beijing may be a new finding, although the interannual relationship has been revealed by many previous studies. As you argued, the relationship between NAO and haze is stronger after mid-1990s. Thus, can you composted this relationships using the observations of PM2.5 concentrations? As you known, the PM2.5 concentrations were widely observed since 2014.

**Response:** *Many thanks for this suggestion. However, there are two reasons why we do not use the PM2.5 concentration in our study. The first is that our study is conducted for the 37-year period of 1980-2016, while quality ground-based PM2.5 data is only available from 2013 (see Xue, W. et al., 2021: Spatiotemporal PM2.5 variations and its response to the industrial structure from 2000 to 2018 in the Beijing-Tianjin-Hebei region. Journal of Cleaner Production, **279**, 123742, doi: 10.1016/j.jclepro.2020.123742). Hence these data would overlap our interval for only four years*

*(2013-2016). We do not believe that statistically robust conclusions could be reached with this short period. Xue et al. also made use of high resolution PM2.5 data which was derived via a regression algorithm applied to MODIS MAIAC satellite data. This set extended back to 2000 (and hence covered only half the period of interest here). Xie et al. also commented that the set needs to be treated with caution '... because it is limited by the statistical regression model, the estimation and prediction ability of the model still has some deviation, which may lead to certain differences in historical PM2.5 concentrations between satellite observations and the real situation'.*

*Secondly, we mainly consider the meteorological definition of haze which is defined by relative humidity and visibility. These two meteorological elements are more directly related to atmospheric circulations.*

Data and Methods: The details of composite approach should be introduced. For example, how to determine day -10, day 0 and day 4 of a persistent haze events lasting for more than 5 days? and so on……

**Response:** *Thanks for pointing out the need to introduce this more carefully. We only mentioned "..., where Day 0 denotes the day with the minimum visibility within a persistent haze event". However, we did not make it very clear and mention it in the Data and Method Section. In the revised manuscript, we have added more details about this approach at the end of Section 2.2.*

Line 197–207: What is the differences between the definition of NAO and your LW03 NAO? It should be carefully illustrated and marked on Figures. Furthermore, I cannot understand the large differences of the correlation coefficients when you only change the definition of NAO. If this evident differences existed, the LW03 NAO you defined is still the NAO pattern? or something else?

**Response:** *We used the NAO index of NOAA/CPC and a modified LW03 NAO index in this study. The NAO index of NOAA/CPC is identified by the Rotated Principal Component Analysis (RPCA) technique. The LW03 NAO index is the difference of zonally-averaged (from 80°W to 30°E) SLP between 35°N and 65°N. One difference of our modified NAO index from the LW03 NAO index is that we do the zonal average from 80°W to 10°W. In the revised manuscript, we name this modified NAO index the Western-type NAO (WNAO) index. This NAO index better reflects the amplitude of a north-south dipole over the North Atlantic, which corresponds to the circulation pattern of the Beijing haze. Thus, it has a higher correlation with the haze frequency. Following the studies of Yao and Luo (2014) and Luo et al. (2014), we also call it a NAO pattern, but a more westward NAO pattern, the WNAO pattern. We have added more explanatios as to why we define this WNAO index in section 2.1.*

Line 207: I did not agreed that the connection between haze and EA/WR pattern is weaker. The correlation coefficients, listed from Line 205 to Line 210, have little difference. In addition, Line 210–219 cannot stand, as you showed decadal changes of connections between NAO/EAWR and haze.

**Response:** *Thanks for picking these things up. In the descriptions of the correlations between the winter haze days and the atmospheric teleconnections, we have changed 'NAO index' into 'WNAO*

*index', since what we focused on is a western type NAO in this study.*

*Compared with the NOAA/CPC EA/WR index (0.36, p<0.05), the NOAA/CPC NAO index (0.27, p<0.1) has a weaker correlation with the winter haze days over the time period 1980-2016. However, our modified WNAO index has a higher correlation (0.42, p<0.01) with the winter haze days, that is, 0.06 higher than that of the EA/WR index (and a higher confidence level. We agree that to say 'the connection between haze and EA/WR pattern is weaker' is not accurate. Thus, in the revised manuscript, we have stressed out that it is the western type NAO that has a higher correlation with the winter haze days.*

*Moreover, over the time period 1980-1999, the correlation coefficient for EA/WR index (0.57, p<0.01) is much higher than that of the WNAO index (0.08), while over the time period 2000-2016, the correlation for EA/WR index (0.22) is much lower than that of the WNAO index (0.61, p<0.01). Both of these teleconnection patterns are essential in inducing the winter hazes, but they act on different time periods. In the revised manuscript, we now present a more objective description to show the comparison of their relationships with the winter haze days.*

*On the Lines 210–219, our purpose was to show that the WNAO index better corresponds with years with high and low haze days. Please note that we have removed 'the modified LW03 NAO index is better correlated with winter haze days in Beijing' in Line 218.*

Section 3.2: the discussion about intraseasonal relationship needed rewrite to be more compact and clearer.

**Response:** *Thanks, we appreciate the point you are making here. In the revised manuscript, we have stressed the daily-to-weekly timescales instead of the intraseasonal timescale. We have rewritten the daily-to-weekly relationship in the conclusions and discussions. To make it more compact and clearer, we have removed some detailed descriptions on circulation evolutions and added the prediction part of winter haze.*

It might be better to move discussion about the impacts of sea surface temperature and Arctic sea ice Section Discussion, because you did not fully explained associated physical mechanisms.

**Response:** *Thank you for these very helpful suggestions. In the revised manuscript, we only briefly described the relationships between the SST, SIC and winter haze in Beijing in conclusions. The impacts of SST and SIC in the discussion have been removed. However, since SST and SIC are external forces for atmospheric circulations, we only added some further insights including SSTs and SICs.*

Line 478: the relationship between the subtropical Western Pacific SST and haze over North-Central North China Plain also contributed to the variation in haze of Beijing, as well as the Strengthening Relationship between Eurasian Snow Cover and December Haze Days in Central North China after the Mid-1990s.

**Response:** *Thank you for your comment. We have cited these two papers and they have enriched our literature research.*

Most of the Figure must be improved. Especially, the country boundaries should be carefully examined.

**Response:** *Thank you for this feedback on our graphics. We have improved most of the Figures and checked the country boundaries.*

The language must be improved by native speaker.

**Response:** *Thank you very much, we have asked a native English speaker to help us revise the language of the article.*

References:

Luo D., Yao, Y., and Feldstein, S.: Regime transition of the North Atlantic Oscillation and the extreme cold event over Europe in January-February 2012, Mon. Wea. Rev., 142, 4735–4757, doi:10.1175/MWR-D-13-00234.1, 2014.

Yao Y., and Luo, D.: Relationship between zonal position of the North Atlantic Oscillation and Euro-Atlantic blocking events and its possible effect on the weather over Europe. Sci. China. Earth. Sci., 57, 2628–2636, doi:10.1007/s11430-014-4949-6, 2014.

Xue, W. et al.: Spatiotemporal PM2.5 variations and its response to the industrial structure from 2000 to 2018 in the Beijing-Tianjin-Hebei region. Journal of Cleaner Production, 279, 123742, doi: 10.1016/j.jclepro.2020.123742, 2021.

Yin, Z., and Wang, H.: The relationship between the subtropical Western Pacific SST and haze over North-Central North China Plain, Int. J. Climatol., 3479–3491, doi:10.1002/joc.4570, 2016.

Yin, Z., and Wang, H.: The strengthening relationship between Eurasian snow cover and December haze days in central North China after the mid-1990s, Atmos. Chem. Phys., 18, 4753–4763, doi: 10.5194/acp-18-4753-2018, 2018.

---

## Author Comment (AC2)

Review for Atmospheric Chemistry and Physics manuscript acp-2020-823, "Atmospheric transmission patterns which promote persistent winter haze over Beijing," by Li et al.

This paper analyzes the local and large-scale meteorological conditions that are conducive to haze conditions in Beijing, China. The paper shows that there are physical links between the North Atlantic Oscillation and haze in Beijing, mediated by atmospheric wave patterns that also interact with sea surface temperatures globally and sea ice concentrations in the Arctic.

The paper is written well, the results are clearly presented, and the physical mechanisms proposed are plausible and reasonable. In general, I am supportive of the paper being published. However, I think some revisions are needed in order to justify publication in ACP.

**Response:** *Many thanks for your encouraging comments. We have revised the manuscript according to these comments, and our point by point responses are listed in detail below.*

Major comments:
**Novelty**
My main concern with the manuscript as written is the novelty. The manuscript presents an extensive literature review (lines 40-99) that nicely lays out what we know about haze in Beijing and the meteorology that affects it, but I come away from this section wondering: if we know so much already, what is the contribution of this particular manuscript? The authors say they're interested in the predictability of Beijing haze via the NAO (lines 103-107), but if there are already studies that have illustrated the linkages between the NAO and Beijing haze (lines 59-62), what is new in this study?

My best sense of the novelty of this paper is the notion of predictability. That is, we know that the NAO affects the circulation patterns that are conducive to haze, but does this knowledge give us meaningful predictive power a certain number of days ahead of time? The authors suggest that this may be true at many points, but they do not actually quantitatively test this proposition. Figure 10 is the closest the authors come to this idea, but it's the last figure in the paper and it feels like it's under-developed.

To my mind, the best way to test the idea of predictability here would be to examine the conditional probabilities of haze events given NAO conditions. That is, what is the baseline/average probability of haze and what is the probability of haze given that the NAO is positive (above some threshold), etc.? Does the probability of haze substantially increase given NAO+ conditions relative to the baseline/average probability of haze, and is this increase statistically meaningful? This could be done separately for a set of lead times (10 days, then 9 days, then 8 days, etc.).

The point of this analysis would be to ask: If you are a decisionmaker at day X, and you know nothing about the NAO, what is your baseline expectation for the probability of haze on day X+10? Then, if you do know the state of the NAO, how does your expectation for the probability of haze on day X+10 change? The authors could then test the statistical significance of this change with a bootstrap test, a Monte Carlo simulation, or something similar. Figure 10 is the beginning of this type of analysis, but it's done on interannual timescales, not daily/weekly, and doesn't have a robust statistical treatment.

**Response:** *We appreciate these comments and are grateful about your idea of stressing out the predictability. We have calculated the probability of haze on Day X+10 under different state of NAO following your suggestions. The results are interesting and very encouraging. This has helped to improve the manuscript and added much to novelty.*

**Organization**

My other major comment pertains to the organization of the manuscript. Given the comments I made above, I think there are parts of section 5 that are not necessary or helpful. The paragraphs relating to figures 8 and 9 are somewhat interesting, but do not really add much to the analysis. For example, I am not really persuaded that "Compared with the evolution of the wave train in Fig. 9a, that in Fig. 9b has a longer wavelength, but it is less organized and less persistent. This suggests that the preceding NAO+ pattern could increase the occurrence…" The difference in wave train organization between NAO+ and NAO- conditions is not quantified and difficult to get a handle on. The analysis pertaining to figure 10 is really where things start to get good, and so I think the authors should foreground that type of analysis much more strongly.

**Response:** *Thanks for these observations. We agree that the comparison of the wave train organization between NAO+ and NAO- conditions is difficult to get a handle on and is not much related to this study. In the revised manuscript, we have removed Figure 9 and left Figure 8 to the end. Instead, we have added a penetrating analysis starting from Figure 10, focusing on the predictability of haze under different NAO states.*

The other organization issue that confuses me is the structure of sections 3, 4, and 5. Section 3.1 is focused on interannual timescales. The authors then transition to daily timescales in section 3.2 and then back to interannual timescales in section 4. Then, section 5 discusses the NAO on daily timescales in figure 8 and 9, and then moves back to interannual timescales for figure 10. This seems odd given that the predictability arguments, which seem to be the most novel parts of the manuscript, are primarily relevant on daily-to-weekly timescales.

**Response:** *In response to your comments in the revised manuscript we have focused our research on three aspects: the atmospheric circulation, haze prediction under*

*different NAO states and external influences. In addition, we have explained these three aspects on both interannual and daily-to-weekly timescales, respectively. After a reorganization of whole manuscript, the structure is now much clearer and more organized.*

I recommend the authors think more deeply about which timescale they would like to focus on, given the parts of the manuscript that are the most novel, and describe more explicitly how the sections of the manuscript connect to one another. Some sections may need to be moved around.

**Response:** *Thank you for your comments – Reviewer #1 also made some similar comments on how the paper would benefit from some of the sections being moved around. We think that both the interannual and daily-to-weekly timescales are important in this study and we give equal attention to these two timescales. The interannual timescale analysis presents implications for a longer time prediction of haze, while the daily-to-weekly timescale analysis suggests its significance in the forecast of haze. We have placed the prediction part in the middle of this manuscript and the external influences at the end of this manuscript to distinguish between priorities.*

I emphasize that the pieces are present in this paper for a nice contribution to the literature. I believe all of my comments can be addressed if the authors invest time in the revision.

**Response:** *Thanks for your heartening remarks.*

**Minor comments:**

Lines 12 and 27: When you say "significant," do you mean "statistically significant" or something like "physically meaningful"? I would generally avoid using the word "significant" outside the narrow context of statistical significance, since it can be confusing. Other words you could use here include "pronounced," "anomalous," etc.

**Response:** *Thanks for pointing out the poor wording here. In the revised manuscript, we have corrected the expression as you suggested in the Abstract.*

Line 37: Define PM2.5 before using the acronym ("particulate matter with a diameter of less than 2.5 microns")

**Response:** *Thank you for noting our omission here. In the revised manuscript, we have described the definition of PM2.5 in the first paragraph of section 1.*

Line 59: A weakened E. Asian trough and Siberian high likely also suppress horizontal

diffusion of pollutants, not just vertical diffusion

**Response:** *Thanks. We totally agree that E. Asian trough and Siberian high likely also suppress horizontal diffusion of pollutants and we have added it to the second paragraph of section 1.*

Line 66: Should this read "increased relative humidity," rather than decreased?

**Response:** *Thank you for spotting the mistake here. We have changed 'increased' into 'decreased' in the revised manuscript.*

Lines 140-151: The authors define haze using relative humidity and visibility. Why not define it using certain thresholds of PM2.5 concentrations, since PM2.5 is the primary health hazard associated with haze? (For example: Callahan and Mankin, 2020, Cai et al., 2017.) If it's because PM2.5 data is not available for a long time period, I think the authors should at least mention this.

**Response:** *Thanks for raising this point – a similar one was raised by Reviewer #1. We did not use the PM2.5 data to define haze mainly because of two reasons. The first one is what you mentioned that 'PM2.5 data is not available for a long time period', the second one is that we prefer to use what WMO suggested (relative humidity and visibility) to define haze because we focus more on meteorology than the environmental pollution perspective. In the revised manuscript, we also explained the reasons why we did not use certain thresholds of PM2.5 concentrations in the first paragraph of section 2.2.*

Line 157: I would remove the word "associated" or say "large-scale atmospheric circulations associated with haze". The word "associated" on its own doesn't mean much.

**Response:** *Thank you for your suggestion. You are right. The word 'associated' here is redundant. 'We have removed the word 'associated' in the revised manuscript.*

Line 162: "Significantly different from zero" at what alpha? 0.05, 0.01, etc.?

**Response:** *Thank you for your comment, to signify the significance level, we have added (p = 0.11) to this sentence in the first paragraph of section 3.1.*

Lines 273-292: This paragraph/analysis (corresponding to figure 5) does not feel particularly novel to me. Many, many previous studies have described the local meteorology conducive to haze in Beijing. The presence of the anticyclonic circulation pattern over north China is a key factor that shapes many of these local factors (Callahan and Mankin, 2020, Zhong et al., 2019), and you've already identified the teleconnection between the NAO and that anticyclonic circulation (figure 3) so I think this additional

analysis and figure 5 could be cut.

**Response:** *We agree with your point here. In the revised manuscript, we have removed Figure 5 and related analyses.*

Line 370: I think this should read "blue line," not "green line" (there is no green line in figure 8).

**Response:** *Thank you for pointing out this mistake, we have corrected this in the second paragraph of section 5.2.*

Figure 2: I have a hard time judging correlations from time series plots. It would be nice to see scatter plots with each climate index on the x-axis and haze days on the y-axis. The scatter plots could be placed to the right of each timeseries to make a two-column figure. The authors could also point out specific extreme years mentioned in the text (e.g., 2013) in the scatter plots by coloring the points differently or something like that.

**Response:** *Thanks for your constructive comment. We have added the scatter plots as you suggested in Figure 2. The scatter plots present us with a clearer view of the extreme years.*

References:

Cai, W., Li, K., Liao, H., Wang, H., & Wu, L. (2017) Weather conditions conducive to Beijing severe haze more frequent under climate change. Nature Climate Change, 10.1038/nclimate3249

Callahan, C.W. & Mankin, J.S. (2020) The influence of internal climate variability on projections of synoptically driven Beijing haze. Geophysical Research Letters, 10.1029/2020GL088548

Zhong, W., Yin, Z., & Wang, H. (2019) The relationship between the anticyclonic anomalies in Northeast Asia and severe haze in the Beijing-Tianjin-Hebei region. Atmospheric Chemistry and Physics, 10.5194/acp-19-5941-2019